# COVID-19 vaccine hesitancy among adults in Liberia, April–May 2021

**Lily M. Sanvee-Blebo**[1]\*, **Peter A. Adewuyi**[1], **Faith K. Whesseh**[1], **Obafemi Joseph Babalola**[1], **Himiede W. Wilson-Sesay**[1], **Godwin E. Akpan**[1], **Chukwuma David Umeokonkwo**[1], **Peter Clement**[2], **Maame Amo-Addae**[1]

1 Liberia Field Epidemiology Training Program, Monrovia, Liberia, 2 World Health Organization, Monrovia, Liberia

\* lblebo@afenet.net

## Abstract

### Background

Vaccination is one of the most cost-effective public health interventions used to prevent diseases in susceptible populations. Despite the established efficacy of vaccines, there are many reasons people are hesitant about vaccination, and these reasons could be complex. This rapid survey estimated the prevalence of COVID-19 vaccine hesitancy and potentially contributing factors in Montserrado and Nimba counties in Liberia.

### Methods

A cross-sectional study was conducted among adults living in Liberia. The relationship between vaccine non-acceptance and sociodemographic characteristics was examined using chi-square statistics. The variables with a p-value less than 0.2 at the bivariate analysis were modelled in a binary logistic regression at a 5% level of significance. The adjusted odds ratio and 95% confidence interval are reported.

### Results

There were 877 participants in the study. Majority were 25–34 years of age (30.4%, 272/877), females (54.05%, 474/877), and Christians (85.2%, 747/877). Most of the participants were aware of the COVID-19 vaccine (75%, 656/877), single (41.4%, 363/877), self-employed (37.51%, 329/877), and live-in rural communities (56.1%, 492/877). Vaccine hesitancy was (29.1%, 255/877; 95% CI:26.2–32.2). Vaccine hesitancy was greater among adults living in urban areas (41%) compared to persons living in rural communities (59%) (aOR; 1.5, 95% CI: 1.1–2.1) and respondents aged 45–54 years (aOR:0.5; 95% CI: 0.2–0.9; $p$ = 0.043) were 50% less likely to be hesitant to COVID-19 vaccination compared to those more than 55 years. The most common source of information was the media (53%, 492/877) and the main reason for being hesitant was a need for more information about the vaccine and its safety (84%, 215/255).

**Data Availability Statement:** All data are available within the manuscript and its supporting information.

**Funding:** The author(s) received no specific funding for this work.

**Competing interests:** The authors have declared that no competing interests exists.

## Conclusions

The majority of study participants were aware of the COVID-19 vaccines and their most common source of information was the media (television, radio). Vaccine hesitancy was moderate. This could pose a challenge to efforts to control the spread of the COVID–19 pandemic. Therefore, the health authorities should provide more health education on the importance of vaccines and their safety to the populace.

## Background

Vaccinations represent effective public health interventions for communicable diseases. They are cost-effective in protecting populations from exposure and transmission of communicable diseases [1]. Ensuring that the coronavirus disease (COVID-19) vaccine is acceptable and available at the community level has been a key prevention strategy during the current pandemic [2]. It has been estimated that acceptance of COVID-19 vaccinations could prevent 2–3 million deaths a year, and a further 1.5 million new COVID-19 infections could be avoided if global coverage of vaccinations improves [3]. A widely accessible, safe, and acceptable vaccine is essential to mitigating the health and economic impact of the pandemic [4]. By April 8, 2020, more than 100 COVID-19 vaccine candidates were under development [5–7]. Although global development of these vaccines has proceeded at a fast pace, only 2% of global clinical trials for vaccines have taken place in Africa [8]. Currently, more than six COVID-19 vaccines have been approved for human use in many countries but in Liberia, only three vaccines are available, Astra Zeneca, Johnson and Johnson, and Pfizer [9,10]. To support countries, the World Health Organization launched the Strategy to Achieve Global COVID-19 Vaccination by mid-2022 which encourages countries to meet the target to vaccinate 40% of their population by the end of 2021 and 70% by mid-2022 [11]. In Liberia, a high hesitancy of greater than 30% means we may not attain the target, a moderate of 20% - 30% means we can achieve but must make additional effort to rally all the non-hesitant and change mindset of the hesitant, and low of less than 20% means we will achieve the target of COVID -19 coverage.

Despite the established efficacy of vaccines, many people still do not take the vaccine for many reasons, some of which could be complex. Conspiracy theories, myths and misconceptions, and lack of understanding of the importance of vaccination have been reported as reasons for vaccine hesitancy. According to the WHO vaccine advisory group, complacency, inconvenience in accessing COVID-19 vaccinations, and lack of confidence are key reasons underlying vaccine hesitancy [3]. These reasons lead to significant increases in vaccine hesitancy and an associated increase in illness and death from vaccine-preventable diseases, posing large economic costs for healthcare to the society [12]. Understanding the level of and reasons for vaccine hesitancy in the population is important for planning towards improving vaccine acceptance. Many countries have introduced COVID-19 vaccination and are exploring the factors that drive vaccine hesitancy in different population groups. When Liberia launched its nationwide COVID-19 vaccination campaign in April 2021, initial uptake was low; insight into the contributing factors was limited. This rapid survey was conducted to provide information about commonly held perceptions of COVID-19 vaccines and the factors potentially contributing to vaccine hesitancy with the end goal of helping Liberian health authorities increase COVID-19 vaccine coverage.

## Methods

### Study area

Liberia is made up of 15 counties, all of which have reported COVID-19 cases. Of the 2,099 confirmed cases reported in the country as of April 27, 2021, Montserrado County reported 1,538 (75.9%) and Nimba County 71 (3.5%), making them the top two counties reporting confirmed cases. In addition, Montserrado and Nimba Counties are the top two densely populated counties in Liberia, with approximately 1,646,421 (33%) and 462,026 (13%) of the Liberian population respectively [13]. Montserrado is considered a primarily urban county (61% urban communities 1967/3230, seven health districts, 22 zones) while Nimba County is predominantly rural (11% rural communities (159/1471) with six health districts). There are 112 healthcare workers per 100,000 population in Liberia.

Montserrado County contains a four-tier health system (county, district, zonal, and community level). Nimba County, however, has only three reporting tiers (county, district, and community level). Montserrado County has 351 health facilities across seven health districts (most health facilities are privately owned); Nimba County has 75 health facilities in six health districts. Routine immunization services are offered at all levels of the health system. This rapid survey was conducted in Montserrado and Nimba Counties.

### Study design and population

A cross-sectional study was conducted among Liberia´s target group for the COVID-19 vaccine (ages 18 years or above). Adults who declined to participate and persons too sick to appropriately respond to the interview were excluded.

### Sample size

The sample size was calculated using the Kish and Leslie formulae $n = Z_{\alpha/2}^2 * p*(1-p)/d^2$ with the following assumptions, Z = 1.96, p proportion of vaccine hesitancy rate is 42.2% [14], margin of error at 5%, and design effect of 2 [15]. The minimum sample size was 750, however, with an anticipated 15% non-response rate we aimed for a sample size of 882.

### Sampling technique

A multi-stage sampling technique was used to recruit the participants. **Stage 1:** we obtained the list of communities in the two counties selected for the study. The two counties have a total of 4,701 communities. The communities were stratified into urban and rural. Montserrado has 1,967 urban and 1,263 rural communities; Nimba has 159 urban and 1,312 rural communities. From each stratum, communities were selected using simple random sampling. The number of communities selected was proportionate number of communities in each stratum in each county. Twenty-five communities were selected in all. Finally, seventeen (seven rural and ten urban) communities were selected in Montserrado; and eight (seven rural and one urban) were selected in Nimba using simple random sampling.

**Stage 2:** In each community selected during stage 1, a sample size of 35 respondents was allocated. A systematic sampling technique was used to select households within each community. The sampling interval was obtained by dividing the number of households in the community by the sample size allocated to the community. The sampling in each community started from a major landmark in the community such as a market square, a community leader's house, and a religious center. The data collector spun a bottle/pen to show the direction to follow. The first household was selected from the community landmark by simple random sampling within the first sampling interval. The subsequent households were selected based on the

sampling interval until the sample size was obtained. The interval varied from one community to the other based on number of households in each community.

**Stage 3:** In each selected household, all the eligible members of the household were line listed and one eligible member of the selected household was selected by balloting to participate. If a household do not have an eligible respondent, the next systematically selected household was identified until the sample size was obtained.

## Data collection

Data was collected with a structured questionnaire (S1 File) adapted from a previous study on the vaccine acceptance [2,16–21]. The questionnaire collected data on respondents' socio-demographic characteristics, sources of information on the COVID-19 vaccine, and attitudes towards the uptake of the vaccine. Section A of the questionnaire captured participants' demographic characteristics including age, sex, marital status, occupation, educational status, religion, and area of residence. Participants' knowledge and practice of COVID-19 prevention were captured in section B, while section C focused on the attitude (perception and acceptance) towards the COVID-19 vaccine.

Responses from the questionnaires were captured electronically using an android device. Each trained interviewer obtained informed consent before the start of data collection.

## Training of data collector and field staff

A one-day training was conducted for data collectors in each county. These data collectors were graduates of Liberia's Field Epidemiology Training Program (LFETP) who worked in the communities selected for the survey or Liberia Demographic and Health Survey field team members. The training covered research ethics and informed consent, questionnaire review, and understanding of the data collection tools.

The questionnaire was pre-tested in a community not selected for the survey and corrections were made before final deployment to the field. The data were collected by 25 data collectors and was supervised by 7 supervisors (3 in Montserrado and 4 in Nimba). More supervisors were recruited in Nimba county because of the greater proportion of geographically distanced "hard-to-reach" rural areas. The supervisors ensured compliance with the protocol and supported data collectors in navigating challenges to ensure the collection of useful data.

## Data analysis and management

The data (S2 File) were cleaned and analyzed using Epi Info version 7.2. Sociodemographic characteristics were presented in frequencies and proportions. Vaccine acceptance was defined as the proportion of the participants who would accept or who had already taken the COVID-19 vaccine while those who would not or had not accepted the vaccine were categorized as hesitant. The relationship between COVID-19 vaccine hesitancy and sociodemographic characteristics was examined using chi-square statistics. The variables with a probability value (*p*-value) less than 0.2 at the bivariate analysis were modelled in a binary logistic regression at a significance level of 5%. The adjusted odds ratio (aOR) and 95% confidence interval (CI) were reported. Reasons given for vaccine hesitancy were grouped by common themes.

## Ethical clearance

Written informed consent (in S1 File) was obtained from the participants after informing them of the risks and benefits of their participation. The study was approved by the Ministry

of Health through the National Incident Management System (IMS) for COVID-19 response (S3 File). The leadership of the Ministry of Health and the National Public Health Institute of Liberia constitute the IMS [22]. The personal identifiable information was not collected from the participants and the confidentiality of the data was maintained by ensuring only study team members with appropriate clearance had to access to the data for research purposes only.

## Results

A total of 877 respondents participated in the study, yielding a response rate of 99.4%. The mean age of the respondents was (38±13 years). The highest was 25–34 years of age (30.4%, 267/877), females (54.1%, 474/877), and unmarried (41.4%, 363/ 877). Most of the participants were employed (60%, 526/877), Christian (85.2%, 747/877), and lived in rural communities (56.1%, 492/ 877) (Table 1). Awareness of the COVID-19 vaccine among the participants was (75%, 656/877) and the COVID-19 vaccine hesitancy rate among the respondents was (29.1% 255/877, 95% CI: 26.2–32.2).

**Table 1. Sociodemographic characteristics of respondents, Montserrado and Nimba, 2021.**

| Respondent Characteristics | Frequency (N = 877) | Percentage (%) |
|---|---|---|
| **Sex** | | |
| Male | 403 | 45.9 |
| Female | 474 | 54.1 |
| **Age group (years)** | | |
| 15–24 | 134 | 15.3 |
| 25–34 | 267 | 30.4 |
| 35–44 | 225 | 25.7 |
| 45–54 | 142 | 16.2 |
| ≥ 55 | 109 | 12.4 |
| **Religion** | | |
| Christianity | 747 | 85.2 |
| Islam | 116 | 13.2 |
| Traditional | 14 | 1.6 |
| **Education** | | |
| No formal education | 151 | 17.2 |
| Primary | 70 | 8.0 |
| Secondary | 422 | 48.1 |
| Tertiary | 234 | 26.7 |
| **Marital Status** (N = 874) | | |
| Unmarried | 363 | 41.5 |
| Married | 333 | 38.1 |
| Other* | 42 | 4.8 |
| Co-habiting | 136 | 15.6 |
| **Employment status** | | |
| Employed | 526 | 60.0 |
| Unemployed | 351 | 40.0 |
| **Town Type** | | |
| Rural | 492 | 56.1 |
| Urban | 385 | 43.9 |

*Divorced, widow/widower, separated

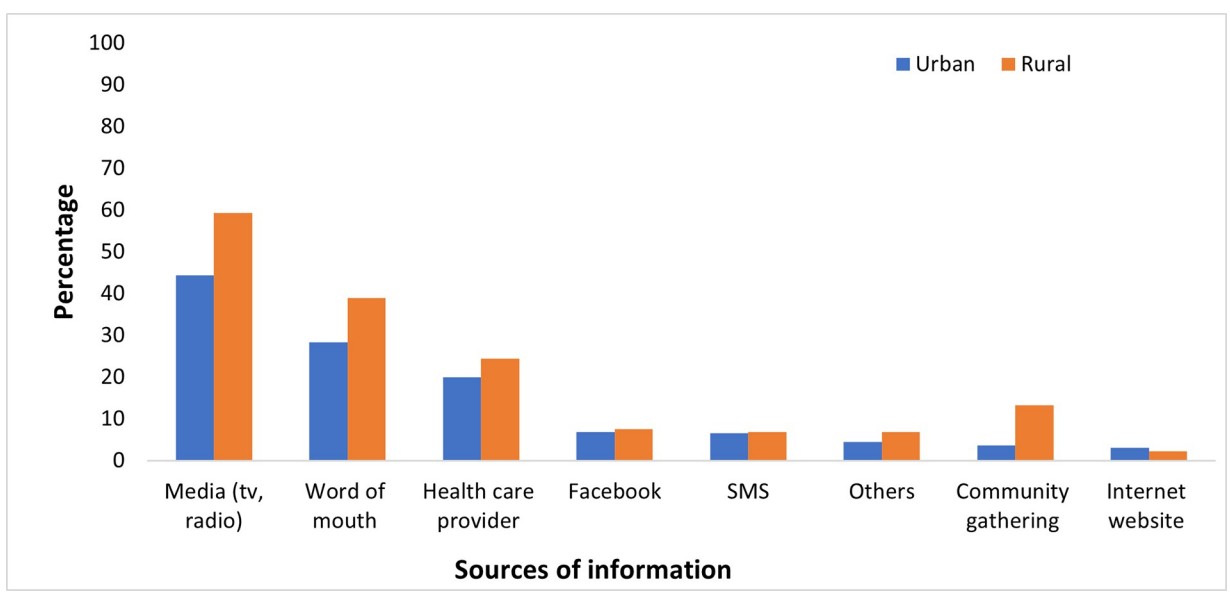

**Fig 1. Respondent source of information by town type, Montserrado and Nimba, 2021.** Note: Respondents may have provided multiple answers. "Other" includes Twitter, WhatsApp, parents & caregiver, meetings.

The most common source of information about COVID-19 and the COVID-19 vaccine was broadcast media (53%, 463/877) while the least common was Short Message Service (7%, 67/877) and internet-based platforms (6%, 59/877) (Fig 1). The most commonly reported reason for vaccine hesitancy was the need for more information (84%, 214/255), the second most common reason was concerns about safety and concern for side effects (12% 31/255), and the third most common reason was conspiracy theories (4%, 10/255) (Fig 2).

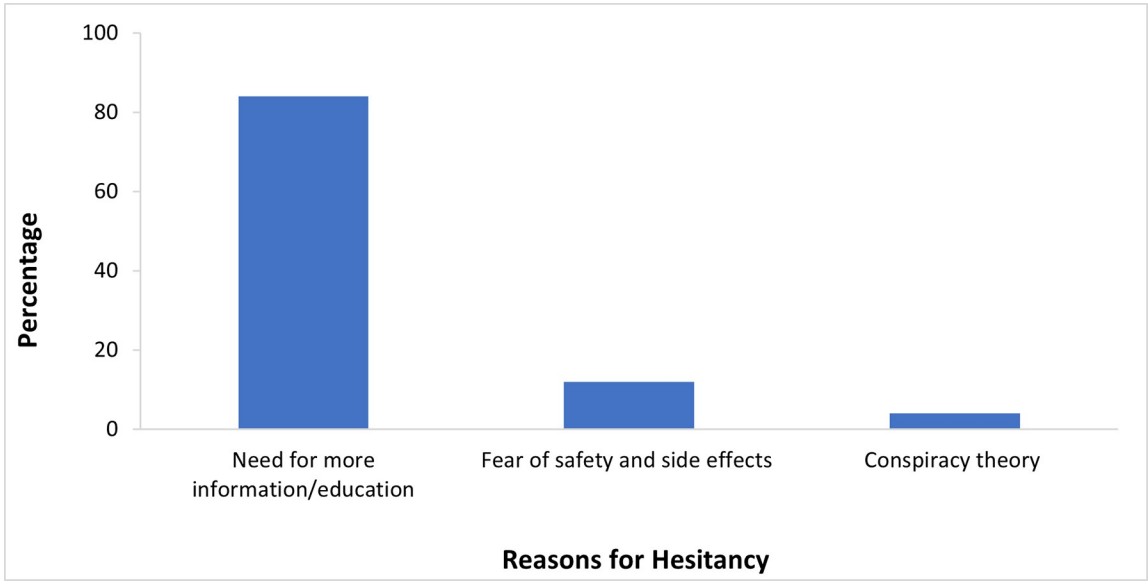

**Fig 2. Reasons for COVID-19 vaccine hesitancy among respondents, Liberia, 2021.**

**Table 2. Bivariate and multivariable association between sociodemographic characteristics and respondent's intention of COVID -19 vaccine, Liberia, 2021.**

| Variable | Hesitancy | | OR | aOR (CI) | p-value* |
|---|---|---|---|---|---|
| | **Yes** | **No** | | | |
| **Sex** | | | | | |
| Female | 151 (31.9) | 323 (68.1) | 1.3 (1.0–1.8) | 1.4(1.0–2.0) | 0.061 |
| Male | 103 (25.7) | 298 (74.3) | 1 | 1 | |
| **Age group (years)** | | | | | |
| 15–24 | 51 (38.1) | 83 (61.9) | 1.3 (0.8–2.3) | 0.98 (0.5–1.8) | 0.954 |
| 25–34 | 77 (28.8) | 190 (71.2) | 0.9 (0.5–1.5) | 0.75 (0.4–1.3) | 0.335 |
| 35–44 | 64 (28.4) | 161 (71.6) | 0.9 (0.5–1.5) | 0.78 (0.4–1.3) | 0.397 |
| 45–54 | 30 (21.1) | 112 (78.9) | 0.6 (0.3–1.1) | 0.5 (0.2–0.9) | 0.043 |
| ≥55 | 33 (30.1) | 75 (69.4) | 1 | 1 | |
| **Education** | | | | | |
| No formal education | 48 (31.8) | 103 (68.2) | 1 | 1 | |
| Primary | 16 (23.2) | 53 (76.8) | 0.6 (0.3–1.2) | 0.6 (0.3–1.3) | 0.201 |
| Secondary | 130 (30.8) | 292 (69.2) | 0.9 (0.6–1.4) | 0.8 (0.5–1.3) | 0.410 |
| Tertiary | 60 (25.8) | 173 (74.3) | 0.7 (0.4–1.2) | 0.6 (0.4–1.0) | 0.068 |
| **Marital Status** | | | | | |
| Unmarried | 125 (34.4) | 238 (65.6) | 1.7 (1.2–2.8) | 1.6 (0.9–2.6) | 0.059 |
| Married | 84 (25.2) | 249 (74.8) | 1.1 (0.7–1.8) | 1.2 (0.7–1.9) | 0.589 |
| Other** | 15 (34.4) | 30 (66.6) | 1.6 (0.8–3.5) | 1.3 (0.6–3.1) | 0.463 |
| Co-habiting | 31 (22.8) | 105 (77.2) | 1 | 1 | |
| **Town Type** | | | | | |
| Rural | 123 (25) | 369 (75) | 1 | 1 | |
| Urban | 132 (34.3) | 253 (65.7) | 1.5 (1.2–2.1) | 1.5 (1.1–2.1) | 0.011 |
| **County** | | | | | |
| Montserrado | 188 (32) | 407(68) | 1.4 (1.1–2.0) | 1.1(0.8–1.7) | |
| Nimba | 67 (26) | 215 (74) | 1 | 1 | 0.692 |
| **Religion** | | | | | |
| Christianity | 218 (29.2) | 529 (70.8) | 1 | - | - |
| Islam | 33 (28.4) | 83 (71.6) | 0.9 (0.6–1.5) | - | - |
| Traditional | 4 (28.6) | 10 (71.4) | 0.9 (0.3–3.1) | - | - |
| **Employment status** | | | | | |
| Employed | 147 (27.9) | 379 (72.1) | 1 | - | - |
| Unemployed | 108 (30.8) | 243 (69.2) | 1.1 (0.8–1.5) | - | - |

\* P value reported for adjusted analysis only

\*\* Other: divorced, widow/widower, separated

aOR: adjusted odds ratio; OR: odds ratio

While the majority of the respondents were aware that the COVID-19 vaccine was available in Liberia (75%, 656/873), only 10.6% (51/873) reported knowing someone who had taken the vaccine. The age group 15–24 years old, female gender, unmarried, living in urban areas, and residing in Montserrado County were found to be associated with COVID-19 vaccine hesitancy in bivariate analysis (Table 2). When considering the effect of other variables in the model, respondents aged 45–54 years (aOR:0.5; 95% CI: 0.2–0.9; $p$ = 0.043) were 50% less likely to be hesitant to COVID-19 vaccination compared to those more than 55 years. Those living in urban areas (aOR: 1.5; 95%CI: 1.1–2.1; $p$ = 0.011) were 1.5 times more likely to be COVID-19 vaccination hesitant compared to those in rural areas (Table 2).

## Discussion

A month after Liberia launched its COVID-19 vaccination campaign, findings from this study indicate that COVID-19 vaccine hesitancy was moderate. About one in three adults among the study population reported they would not accept the COVID-19 vaccine. This could be a challenge to achieving high vaccine coverage and overall control of the pandemic, especially with the new more virulent strains of the virus. Health officials responsible for deploying COVID-19 vaccination to the populace need to bear this in mind and develop appropriate interventions to improve acceptance of the vaccine. Though vaccine hesitancy in other low-resource countries appears comparable to our findings [1], a few high-resource countries have also reported similar rates [8,23]. The trend of the pandemic is rapidly changing as more information is provided to the populace and as more people are in a geographic area where the vaccine has become available [24].

The study also found that vaccine hesitancy was not affected by the gender of an individual. In comparison to a study in Nigeria assessing COVID-19 vaccine hesitancy for six geopolitical zones also supports our findings [25].

The finding that television and radio were the most common sources of information followed by word of mouth should be taken into consideration when developing risk communication strategies. Radio and television serve important roles in providing critical information about the pandemic and should be strengthened. It was, however, surprising that despite improvement in mobile phone penetration within Liberia (up to 83% in 2020), this modality did not appear to be a commonly used platform for the communication of pandemic information. The policymakers in health education and communication should consider reviewing current strategies for reaching the populace with important public health information.

A major reason provided by the hesitant respondents for vaccine hesitancy was the perceived need for more information or education about the vaccine. Lack of knowledge has been found as a major factor influencing human behavior towards public health interventions, risk perception, and preventive practices [26]. There is an urgent need to review the type of information being released concerning the vaccine, the channel of communication and its effectiveness. The rapid spread of false information and higher social media usage could explain why vaccine hesitancy was higher in the urban compared to the rural areas. In contrast, the main sources of communication in the rural areas are local radio stations which are in every county, even the hard-to-reach communities. Health care providers were the third most common source of information for both the rural and urban communities. In other published literature, Earnshaw *et al.* reported that healthcare providers were the most trusted source of information about COVID-19 vaccines [27]. In previous studies conducted in Liberia, an increase in mobile phone use has facilitated an increase in access to social media platforms. Other studies demonstrated that social media platforms as sources of information are associated with more doubts and misbeliefs regarding the COVID-19 vaccine [28]. This might be related to the ease of disseminating false information observed on social media, including the possible spread of incorrect safety information about COVID-19 vaccine safety.

Major reasons for non-acceptance included the need for more information about the vaccine, fear of side effects of the vaccine, and concerns about vaccine safety. This could be because of inadequate community engagement as well as not maximizing the use of trusted media institutions or other available modes of communication. Other reasons for hesitancy were the fear of the vaccine and worry about the side effects of the vaccine which has also been reported [10].

Those aged more than 55 years were more likely to be hesitant about COVID-19 vaccine uptake. This may be due to limited information about the vaccine or some myth about the

vaccine. Being a new disease, the younger individuals who are more on social media are more likely to access more information about the disease and may be more accepting of the vaccine. Other studies have shown that younger age groups were however more hesitant as compared to the elderly [29].

Our multivariable regression findings also suggest that location of residence (urban *vs.* rural) was significantly associated with hesitancy to the COVID-19 vaccine. Persons living in urban communities were less likely to accept the vaccine as compared to individuals living in rural communities. Urban dwellers were usually more likely to have mobile phones and engage in social media and more likely to obtain information from other sources that might lead to them having access to false information about the vaccine.

Our study is not without limitations. A qualitative arm would have helped collect in-depth information to help understand other possible reasons for the hesitancy observed. Being only quantitative in nature, our study was not able to explore these. Also, we were not able to follow up with the participants to determine those who eventually took the vaccine.

## Conclusions

The majority of study participants were aware of COVID-19 vaccines. Their most common source of information was broadcast media (television, radio) though not statistically different. The hesitancy rate was moderate, and this could pose a challenge for efforts to control the spread of the COVID–19 pandemic. The survey provides information needed to strengthen and increase the impact of future planned mass vaccination campaigns.

Liberia's health authorities should conduct more health education on the importance of vaccines, targeting persons living in urban areas, with a special focus on women given their higher rate of reported hesitancy. Health education packages should address evidence of disease presence in Liberia, benefits of the vaccine, fear issues and conspiracy theories to the populace.

## Supporting information

**S1 File. A sample of questionnaire for COVID-19 vaccine hesitancy in Liberia.**
(PDF)

**S2 File. Database of COVID-19 vaccine hesitancy in Liberia, April-May, 2021.**
(XLSX)

**S3 File.**
(PDF)

## Acknowledgments

The authors would like to thank the U.S. Centers for Disease Prevention and Control, World Health Organization, Liberia National Incident Management System, African Field Epidemiology Network (AFENET), Montserrado County Health Team, Nimba County Health Team, Dr. Denise Roth Allen, Dr. Rachel T. Idowu, Dr. Ellen Yard, Vachel H. Lake and data collectors for their support during the survey.

## Author Contributions

**Conceptualization:** Lily M. Sanvee-Blebo, Peter A. Adewuyi, Obafemi Joseph Babalola, Himiede W. Wilson-Sesay, Peter Clement, Maame Amo-Addae.

**Data curation:** Godwin E. Akpan.

**Formal analysis:** Lily M. Sanvee-Blebo, Peter A. Adewuyi, Chukwuma David Umeokonkwo, Maame Amo-Addae.

**Funding acquisition:** Peter Clement.

**Investigation:** Lily M. Sanvee-Blebo, Faith K. Whesseh, Godwin E. Akpan.

**Methodology:** Lily M. Sanvee-Blebo, Peter A. Adewuyi, Obafemi Joseph Babalola, Himiede W. Wilson-Sesay, Maame Amo-Addae.

**Project administration:** Lily M. Sanvee-Blebo, Peter A. Adewuyi, Peter Clement, Maame Amo-Addae.

**Resources:** Maame Amo-Addae.

**Supervision:** Lily M. Sanvee-Blebo, Peter A. Adewuyi, Faith K. Whesseh, Obafemi Joseph Babalola, Godwin E. Akpan, Peter Clement, Maame Amo-Addae.

**Visualization:** Lily M. Sanvee-Blebo, Peter A. Adewuyi, Obafemi Joseph Babalola, Godwin E. Akpan.

**Writing – original draft:** Lily M. Sanvee-Blebo, Peter A. Adewuyi, Obafemi Joseph Babalola, Maame Amo-Addae.

**Writing – review & editing:** Lily M. Sanvee-Blebo, Peter A. Adewuyi, Faith K. Whesseh, Obafemi Joseph Babalola, Himiede W. Wilson-Sesay, Godwin E. Akpan, Chukwuma David Umeokonkwo, Peter Clement, Maame Amo-Addae.

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
