## [Decision Letter · Decision Letter 0]

6 Oct 2022

PONE-D-22-17733COVID-19 Vaccine Hesitancy among adults in Liberia, April – May, 2021PLOS ONE

Dear Dr. Sanvee-Blebo,

Thank you for submitting your manuscript to PLOS ONE. After careful consideration, we feel that it has merit but does not fully meet PLOS ONE’s publication criteria as it currently stands. Therefore, we invite you to submit a revised version of the manuscript that addresses the points raised during the review process.

ACADEMIC EDITOR: The manuscript is relevant. However, there are major areas that need revision.The authors should clarify the authority of the ethical approval institution.The methods section should be improved including the study design, the sampling technique and number of interviewers.The discussion should be limited to scope and focus of the current study.The omitted references should be included in the revised manuscript.Please submit your revised manuscript by Nov 20 2022 11:59PM. If you will need more time than this to complete your revisions, please reply to this message or contact the journal office at plosone@plos.org. Please include the following items when submitting your revised manuscript:A rebuttal letter that responds to each point raised by the academic editor and reviewer(s). You should upload this letter as a separate file labeled 'Response to Reviewers'.A marked-up copy of your manuscript that highlights changes made to the original version. You should upload this as a separate file labeled 'Revised Manuscript with Track Changes'.An unmarked version of your revised paper without tracked changes. You should upload this as a separate file labeled 'Manuscript'.

We look forward to receiving your revised manuscript.

Kind regards,

Martin Nyaaba Adokiya, Ph.D

Academic Editor

PLOS ONE

Journal Requirements:

The study was funded by the US. Centers for Disease Control and Prevention (US. CDC) Liberia and World Health Organization (WHO) Liberia 

Additional Editor Comments:

The reviewers have provided important comments to improve the quality of the manuscript. In addition to these comments, the authors should revise the manuscript according to the guidelines of the journal.

Reviewers' comments:

Reviewer's Responses to Questions

**Comments to the Author**

1. Is the manuscript technically sound, and do the data support the conclusions?

Reviewer #1: Yes

Reviewer #2: Yes

2. Has the statistical analysis been performed appropriately and rigorously? 

Reviewer #1: Yes

Reviewer #2: No

3. Have the authors made all data underlying the findings in their manuscript fully available?

Reviewer #1: Yes

Reviewer #2: Yes

4. Is the manuscript presented in an intelligible fashion and written in standard English?

Reviewer #1: Yes

Reviewer #2: Yes

5. Review Comments to the Author

Reviewer #1: The study provides valuable data that shows the level of COVID-19 vaccine hesitancy in Liberia. The manuscript is well written, and I have no major issues with it. I have just one suggestion for the authors, and I need some clarification on the ethical approval.

1. On page 3, first paragraph under background, the authors indicated that only three COVID-19 vaccines were available in Liberia. I suggest the authors indicate the names of those specific COVID-19 vaccines.

2. The authors indicated that the study was approved by the National Incident Management System for COVID-19 responses, Ministry of Health, Liberia. Is the National Incident Management System for COVID-19 response in Liberia an ethical review committee or board? If not, what was the approval for? Is the US CDC non-research determination clearance an ethical clearance? If not, what was the clearance for?

Reviewer #2: A very important paper that would contribute to understanding the attitude of individuals towards the COVID-19 vaccine uptake. The organization of the manuscript is very good. However, there are essential aspects of the paper that needs to be addressed.

Background:

The background was well written. Authors should change the > (i.e.>30%) and (<20%) symbol to words. Some of the statements require citation(s) as the information is not emanating from the authors.

Methods:

Study design and population.

What is rapid cross-sectional study - “rapid” should be deleted.

Major revision:

The sampling technique requires a major revision.

1. Was the study stratified by rural and urban. This should be clear in the write up.

2. Stage 2. A table indicating the population size of each community and the number sample from each community will aid easy understanding of what was actually done.

3. The sampling interval should be stated. If it varies by community due to the different population size of the communities, this should be indicated clearly.

4. Stage 3: looks confusing. What is the relevance of stage 3 to this current paper?

5. The section on street interviews can be deleted. It suggests the authors where selecting participants hazardly. It masks the scientific merit of the whole paper. Authors should look at it critically if it does not contribute to the paper, it has to go/deleted. Otherwise, it should be revised. Importantly, how many interviewers where stationed to conduct this street interviews. Are the authors suggesting that they are able to complete an interview before the tenth participant?

Training of data collection

One-day training is mostly considered inadequate. Did the authors pilot the tool with the research assistants?

Data analysis.

Authors did a multi-stage sampling and stratified, therefore an appropriate statistical analysis has to be performed. The study requires a complex data analysis.

Results

When starting a sentence with a figure, it should be in words. Authors should check all that and correct.

Table 2 can be deleted. All information can be found in table 3.

Discussion

The discussion needs major revision. The paper was poorly discussed. There was overelaboration of issues or study findings. The authors should consider the design of the study and its limitation. Moreover, authors make statements that was outside the scope of this paper and the study findings does not support. For example, “The distribution of ownership of radio or television could be skewed to the middle- and high-income earners who could afford it”. Did the authors check the distribution of radio or television in among their study population? There are number of such statements in the discussion. Authors should focus the discussion and write within the context of the study design and its findings.

6. PLOS authors have the option to publish the peer review history of their article (what does this mean?). If published, this will include your full peer review and any attached files.

Reviewer #1: No

Reviewer #2: **Yes: **Emmanuel K. Nakua

---

## [Author Response · Author response to Decision Letter 0]

16 Dec 2022

Editor’s comment

1. The author should clarify the authority of the ethical approval institution

Response:

Thank you for the thorough review of our manuscript and for useful suggestions to improve the understanding of the work. We have carefully gone through the comments and made modifications as suggested in the work. We have also responded to the comments here for ease of reference to the changes made in the work. We hope that the work is now acceptable for publication and will be available to respond to further clarification, if any. 

2. The methods section should be improved including the study design, the sampling technique and the number of interviewers

Response:

The method section has been improved and appropriate information added

3. The discussion should be limited to scope and focus on the current study

Response:

Thank you, the discussion has been edited 

4. The omitted references should be included in the revised manuscript

Response:

The omitted references have been added 

Journal Requirements

Response:

Thank you for the useful links. We have used the links to format our manuscript accordingly.

The study was funded by the US. Centers for Disease Control and Prevention (US. CDC) Liberia and World Health Organization (WHO) Liberia 

Response:

The funding information has been removed from the manuscript and it is stated clearly in the revised cover letter.

Response:

We have updated this part of the manuscript. All data are available within the manuscript and its supporting information.

Response:

Thank you for this observation. We have deleted the ethical statement in the Acknowledgement section.

Reviewer 1 Comments

1. On page 3, first paragraph under background, the authors indicated that only three COVID-19 vaccines were available in Liberia. I suggest the authors indicate the names of those specific COVID-19 vaccines.

Response:

The names of the vaccines have been added to the paragraph in track changes. (Page 3)

2. The authors indicated that the study was approved by the National Incident Management System for COVID-19 responses, Ministry of Health, Liberia. Is the National Incident Management System for COVID-19 response in Liberia an ethical review committee or board? If not, what was the approval for? Is the US CDC non-research determination clearance an ethical clearance? If not, what was the clearance for?

Response:

The Incident Management System of COVID- 19 comprises the leadership of the Ministry of Health (Minister and Deputy Ministers) and the National Public Health Institute of Liberia (Director General). It was the body charged with the responsibility to approve all COVID-19-related studies during the response. Other COVID-19-related studies undertaken and approved by this body during this period have also been published. 

Acquiring the US. CDC clearance is a process that needs to be completed based on the policy of the organization before the study was conducted. However, it is not relevant to the writing of this paper and has been deleted.

Reviewer 2 comments

1. The background was well-written. Authors should change the > (i.e.>30%) and (<20%) symbol to words. Some of the statements require citation(s) as the information is not emanating from the authors.

Response:

The symbols have been changed to words and new citations have been added.

2. Study design and population. What is rapid cross-sectional study - “rapid” should be deleted

Response:

The word “rapid” has been deleted 

3. Was the study stratified by rural and urban. This should be clear in the write up.

Response:

The study was conducted in two of the 15 counties of Liberia. In each county, the communities were stratified into urban and rural communities. The sentence has been rephrased to provide more clarity

4. Stage 2. A table indicating the population size of each community and the number sample from each community will aid easy understanding of what was actually done. 

Response:

The authors appreciate the comments. The stage has been restructured to better communicate what was done. It now reads, “The first household was selected from the community landmark by simple random sampling within the sampling interval. The sampling interval for each community was obtained by dividing the number of households with the sample sizes allocated to the community segment of the study. The interval varied from one community to the other based on number of households in each community.” 

5. The sampling interval should be stated. If it varies by community due to the different population size of the communities, this should be indicated clearly

Response:

As explained above, the sampling interval vary from community to community depending on their population (household) size. However equal number of respondents were sampled from each community.

6. Stage 3: looks confusing. What is the relevance of stage 3 to this current paper?

Response:

The authors appreciate the comment. The study unit of study was the individual not the household. So stage 3 explained how an individual is sampled within the household. It has been edited to better represent that.

7. The section on street interviews can be deleted. It suggests the authors where selecting participants hazardly. It masks the scientific merit of the whole paper. Authors should look at it critically if it does not contribute to the paper, it has to go/deleted. Otherwise, it should be revised. Importantly, how many interviewers where stationed to conduct this street interviews. Are the authors suggesting that they are able to complete an interview before the tenth participant? 

Response:

This section of the work has been deleted as requested.

8. One-day training is mostly considered inadequate. Did the authors pilot the tool with the research assistants?

Response:

The data collectors that collectors were already part of the response team in the field, involved in other trainings conducted previously. The one-day training focused on the protocol and how to select and administer the study tool. There was a pretesting of the study instrument after the training prior to the data collection.

9. Authors did a multi-stage sampling and stratified, therefore an appropriate statistical analysis has to be performed. The study requires a complex data analysis

Response:

The authors agree with the reviewer’s comment on the need for complex data analysis however, our primary indent was not to estimate the prevalence of the hesitancy at subcounty and community level so we did not apply the take the route. The stratification was done to better represent the different segment of the society in the sample.

10. When starting a sentence with a figure, it should be in words. Authors should check all that and correct.

Response:

The corrections have been made in the manuscript.

11. Table 2 can be deleted. All information can be found in table 3.

Response:

The crude odds ratio from Table 2 has been merged to table 3 and the old table 2 deleted.

12. The discussion needs major revision. The paper was poorly discussed. There was overelaboration of issues or study findings. The authors should consider the design of the study and its limitation. Moreover, authors make statements that was outside the scope of this paper and the study findings does not support. For example, “The distribution of ownership of radio or television could be skewed to the middle- and high-income earners who could afford it”. Did the authors check the distribution of radio or television in among their study population? There are number of such statements in the discussion. Authors should focus the discussion and write within the context of the study design and its findings.

Response:

The section has been reviewed and the offending aspects removed.

---

## [Editor Report · Decision Letter 1]

28 Apr 2023

PONE-D-22-17733R1COVID-19 Vaccine Hesitancy among adults in Liberia, April – May 2021PLOS ONE

Dear Dr. Sanvee-Blebo,

Thank you for submitting your manuscript to PLOS ONE. After careful consideration, we feel that it has merit but does not fully meet PLOS ONE’s publication criteria as it currently stands. Therefore, we invite you to submit a revised version of the manuscript that addresses the points raised during the review process.

We look forward to receiving your revised manuscript.

Kind regards,

Palash Chandra Banik, MPhil

Academic Editor

PLOS ONE
---

## [Author Response · Author response to Decision Letter 1]

2 May 2023

Editor’s comment

1. The author should clarify the authority of the ethical approval institution

Response:

Thank you for the thorough review of our manuscript and for useful suggestions to improve the understanding of the work. We have carefully gone through the comments and made modifications as suggested in the work. We have also responded to the comments here for ease of reference to the changes made in the work. We hope that the work is now acceptable for publication and will be available to respond to further clarification, if any. 

2. The methods section should be improved including the study design, the sampling technique and the number of interviewers

Response:

The method section has been improved and appropriate information added

3. The discussion should be limited to scope and focus on the current study

Response:

Thank you, the discussion has been edited 

4. The omitted references should be included in the revised manuscript

Response:

The omitted references have been added 

Journal Requirements

Response:

Thank you for the useful links. We have used the links to format our manuscript accordingly.

The study was funded by the US. Centers for Disease Control and Prevention (US. CDC) Liberia and World Health Organization (WHO) Liberia 

Response:

The funding information has been removed from the manuscript and it is stated clearly in the revised cover letter.

Response:

We have updated this part of the manuscript. All data are available within the manuscript and its supporting information.

Response:

Thank you for this observation. We have deleted the ethical statement in the Acknowledgement section.

Reviewer 1 Comments

1. On page 3, first paragraph under background, the authors indicated that only three COVID-19 vaccines were available in Liberia. I suggest the authors indicate the names of those specific COVID-19 vaccines.

Response:

The names of the vaccines have been added to the paragraph in track changes. (Page 3)

2. The authors indicated that the study was approved by the National Incident Management System for COVID-19 responses, Ministry of Health, Liberia. Is the National Incident Management System for COVID-19 response in Liberia an ethical review committee or board? If not, what was the approval for? Is the US CDC non-research determination clearance an ethical clearance? If not, what was the clearance for?

Response:

The Incident Management System of COVID- 19 comprises the leadership of the Ministry of Health (Minister and Deputy Ministers) and the National Public Health Institute of Liberia (Director General). It was the body charged with the responsibility to approve all COVID-19-related studies during the response. Other COVID-19-related studies undertaken and approved by this body during this period have also been published. 

Acquiring the US. CDC clearance is a process that needs to be completed based on the policy of the organization before the study was conducted. However, it is not relevant to the writing of this paper and has been deleted.

Reviewer 2 comments

1. The background was well-written. Authors should change the > (i.e.>30%) and (<20%) symbol to words. Some of the statements require citation(s) as the information is not emanating from the authors.

Response:

The symbols have been changed to words and new citations have been added.

2. Study design and population. What is rapid cross-sectional study - “rapid” should be deleted

Response:

The word “rapid” has been deleted 

3. Was the study stratified by rural and urban. This should be clear in the write up.

Response:

The study was conducted in two of the 15 counties of Liberia. In each county, the communities were stratified into urban and rural communities. The sentence has been rephrased to provide more clarity

4. Stage 2. A table indicating the population size of each community and the number sample from each community will aid easy understanding of what was actually done. 

Response:

The authors appreciate the comments. The stage has been restructured to better communicate what was done. It now reads, “The first household was selected from the community landmark by simple random sampling within the sampling interval. The sampling interval for each community was obtained by dividing the number of households with the sample sizes allocated to the community segment of the study. The interval varied from one community to the other based on number of households in each community.” 

5. The sampling interval should be stated. If it varies by community due to the different population size of the communities, this should be indicated clearly

Response:

As explained above, the sampling interval vary from community to community depending on their population (household) size. However equal number of respondents were sampled from each community.

6. Stage 3: looks confusing. What is the relevance of stage 3 to this current paper?

Response:

The authors appreciate the comment. The study unit of study was the individual not the household. So stage 3 explained how an individual is sampled within the household. It has been edited to better represent that.

7. The section on street interviews can be deleted. It suggests the authors where selecting participants hazardly. It masks the scientific merit of the whole paper. Authors should look at it critically if it does not contribute to the paper, it has to go/deleted. Otherwise, it should be revised. Importantly, how many interviewers where stationed to conduct this street interviews. Are the authors suggesting that they are able to complete an interview before the tenth participant? 

Response:

This section of the work has been deleted as requested.

8. One-day training is mostly considered inadequate. Did the authors pilot the tool with the research assistants?

Response:

The data collectors that collectors were already part of the response team in the field, involved in other trainings conducted previously. The one-day training focused on the protocol and how to select and administer the study tool. There was a pretesting of the study instrument after the training prior to the data collection.

9. Authors did a multi-stage sampling and stratified, therefore an appropriate statistical analysis has to be performed. The study requires a complex data analysis

Response:

The authors agree with the reviewer’s comment on the need for complex data analysis however, our primary indent was not to estimate the prevalence of the hesitancy at subcounty and community level so we did not apply the take the route. The stratification was done to better represent the different segment of the society in the sample.

10. When starting a sentence with a figure, it should be in words. Authors should check all that and correct.

Response:

The corrections have been made in the manuscript.

11. Table 2 can be deleted. All information can be found in table 3.

Response:

The crude odds ratio from Table 2 has been merged to table 3 and the old table 2 deleted.

12. The discussion needs major revision. The paper was poorly discussed. There was overelaboration of issues or study findings. The authors should consider the design of the study and its limitation. Moreover, authors make statements that was outside the scope of this paper and the study findings does not support. For example, “The distribution of ownership of radio or television could be skewed to the middle- and high-income earners who could afford it”. Did the authors check the distribution of radio or television in among their study population? There are number of such statements in the discussion. Authors should focus the discussion and write within the context of the study design and its findings.

Response:

The section has been reviewed and the offending aspects removed.

---

## [Decision Letter · Decision Letter 2]

10 Jul 2023

PONE-D-22-17733R2COVID-19 Vaccine Hesitancy among adults in Liberia, April – May 2021PLOS ONE

Dear Dr. Sanvee-Blebo,

Thank you for submitting your manuscript to PLOS ONE. After careful consideration, we feel that it has merit but does not fully meet PLOS ONE’s publication criteria as it currently stands. Therefore, we invite you to submit a revised version of the manuscript that addresses the points raised during the review process.

We look forward to receiving your revised manuscript.

Kind regards,

Omar Enzo Santangelo

Academic Editor

PLOS ONE

Reviewers' comments:

Reviewer's Responses to Questions

**Comments to the Author**

1. If the authors have adequately addressed your comments raised in a previous round of review and you feel that this manuscript is now acceptable for publication, you may indicate that here to bypass the “Comments to the Author” section, enter your conflict of interest statement in the “Confidential to Editor” section, and submit your "Accept" recommendation.

Reviewer #3: (No Response)

Reviewer #4: (No Response)

2. Is the manuscript technically sound, and do the data support the conclusions?

Reviewer #3: Yes

Reviewer #4: Partly

3. Has the statistical analysis been performed appropriately and rigorously? 

Reviewer #3: Yes

Reviewer #4: I Don't Know

4. Have the authors made all data underlying the findings in their manuscript fully available?

Reviewer #3: Yes

Reviewer #4: Yes

5. Is the manuscript presented in an intelligible fashion and written in standard English?

Reviewer #3: Yes

Reviewer #4: Yes

6. Review Comments to the Author

Reviewer #3: RE: COVID-19 Vaccine Hesitancy among adults in Liberia, April – May 2021

General comments: This is a well written manuscript and addressed an important issue regarding vaccination in Liberia. The authors have addressed most of the earlier raised issues satisfactorily. However, there are still a few areas that need some improvements.

Abstract

Line 37: “multiple logistic regression” kindly correct as binary logistic regression.

Methods

Lines 102-103: Montserrado is considered a primarily urban county “(61% 1967/3230 seven districts, 22 zones)” this is a bit unclear? What does 1967/3230 indicated? while Nimba County “is predominantly rural (11% (159/1471) six districts)” This is unclear? What does 159/1471 six districts?

Line 119: “The sample size was calculated using the Kish and Leslie formulae (ref).” Kindly insert ref

Line 121: “design effect of 2” what does design effect of 2 meant? Is this part of the formula?

Line 178: “multiple logistic regression” What the authors did was simple binary logistic regression and should be corrected as such.

Results:

Line 200: “SMS” write in full.

Line 210-212: “When considering the effect of other variables in the model, respondents aged more than 55 years (aOR:0.5; 95% CI: 0.2–0.9; p=0.043) were two times more likely to be COVID-19 vaccination hesitant compared to those aged 45–54 years” This statement is at variance and may confuse the reader. It is ether you adjust your reference in the Table for positive odds ratio as reported in the prose or you interpreted as stated in Table 2 in lesser effects.

Line 224: Table 2. “Multivariate association” This is multivariable and not multivariate

Discussion

Lines 271-273: “With regards to age, 15-24 year-olds were more likely to express hesitancy to take the COVID-19 vaccine because the disease is more lethal to persons older and thus they did not see themselves at risk of getting the disease.” Your findings do not support this assertion/discussion. Indeed, it is contrary as vaccine hesitancy was more in those aged more than 55 years! Check your Table 2 results.

Line 276: “statistically significant predictor of the hesitancy of the COVID-19 vaccine” kindly replace with association- a cross sectional data can not give you a predictor. You may wish to read more about predictive models.

Line 278: “Urban dwellers were usually more likely to have mobile phones and engage in social media and more likely to obtain information from other sources” This statement is hanging? Not link with the predecessor and how does it lead to VH?

Line 280: “Our study is not without limitations. Our sample size was small and included only two of Liberia’s fifteen counties. Nevertheless, these counties accounted for 83% of all reported COVID-19 cases in the country” These statements are not limitation. Your sample size of 800 plus is adequately powered and sampling two major districts comprising of urban and rural areas out of 15 should provided an adequate representation and can not be a study limitation. Your study limitations may be lack of in-depth analysis of reasons for VH, which will require a qualitative study approach and also lack of follow up to see how many people will eventually get vaccinated.

Line 282: “The survey provides information needed to strengthen and increase the impact of future planned mass vaccination campaigns” Move to the end of conclusion

Overall while I understood time has caught up with your data, kindly compared your key findings with other African studies e.g Babatunde Oluwatosin Ogunbosi et al. COVID-19 vaccine hesitancy in six geopolitical zones in Nigeria: a cross-sectional survey. Pan African Medical Journal. 2022;42(179). 10.11604/pamj.2022.42.179.34135

Reviewer #4: Thank you for shedding light on an important topic in the Liberian setting. Vaccine hesitancy is complex and while I appreciate your attempt to keep things simple and draw a clear conclusion, I wonder if there would not be a benefit to dealing with some of the nuances in this paper. I will touch on examples of this in the details below. Regarding your conclusion that vaccine hesitancy should be combated through additional information on television and radio since those are the most common sources of information at 53% - this is probably correct but I was not 100% convinced that it was supported by the data presented. Your denominator was the entire study population and I didn't see a breakdown by those who were accepting of vaccination and those who were not.

Other comments by section:

Methods:

Lines 129-130: "...communities were selected based on the number of communities and their distribution in each county." I couldn't really tell from this statement what your selection criteria were. Were you looking for a greater number of communities? What type of distribution did you want, and why?

133-141: I found the description of Stage 2 a little hard to follow and still don't understand how you used the sampling interval to select the second household. Did you randomly select a direction again?

General: I saw a previous reviewer recommended removing the description about street interviews. I am not sure I agree with this if street interviews formed part of your data. The questions I would ask myself are whether they formed a meaningful proportion of your data, and if you can describe it in a way that addresses the comment about it sounding haphazard.

186-187: There was another reviewer who asked about the ethics committee and I also had a question about this. Can you confirm that submitting to the regular ethics committee that you normally would outside a public health emergency was not an option? My concern is that an ethics committee formed of leadership from MOH and NPHIL is not what I would expect in the ERC constitution. For example there may not be diverse backgrounds and no lay person on the committee.

Results:

195: Self-employed is not listed in Table 1, only employed and unemployed.

194-196: To say most participants were unmarried (41.4%) seems a little misleading considering married participants were 38.1% and co-habiting 15.6%.

210-212: Here it says >55 years is more likely to be hesitant than 45-54 but in the abstract you said the opposite.

Table 2: Interesting that the most hesitancy was seen in the 15-24 age group, but this is not mentioned in the results.

Table 2: I think your percentages are reversed for hesitancy yes/no in Montserrado County?

Figure 1: This is where I think it might be helpful to stratify by people who are vaccine hesitant those who are not. Is there a difference between those two groups in sources of information?

Discussion:

254-256: Not sure this is supported by the data? In Figure 1, Facebook as a source of information seems pretty similar between urban and rural.

278-279: Again, I am not seeing this in Figure 1. In fact, it appears that more rural respondents relied on "other" sources of information.

General comments:

As someone who works in vaccine development, I am very interested in the reasons for vaccine hesitancy. As your paper points out, social media is often a source of disinformation. Anti-vax groups exist in echo chambers where information is readily available but the establishment that provides the information is viewed with suspicion. This is in contrast to information bubbles, where there is just a lack of access to information. Your data seem to point to the latter situation in Liberia. However I was interested to note that your survey included questions about whether the government and health authorities are acting in the best interest of Liberians, and which information sources are most trusted. This was not included in the results but could influence how you view the people who were vaccine hesitant. Did you see more distrust in institutions in that group? And is there any new information now that we are a couple years further along, that indicates there is still lack of information?

7. PLOS authors have the option to publish the peer review history of their article (what does this mean?). If published, this will include your full peer review and any attached files.

Reviewer #3: **Yes: **Olayinka Ibrahim

Reviewer #4: No

---

## [Author Response · Author response to Decision Letter 2]

23 Oct 2023

Reviewer # 3 

1 Abstract

Line 37: “multiple logistic regression” kindly correct as binary logistic regression. 

Thanks for your comment, the changes have been made accordingly 

2 Methods

Lines 102-103: Montserrado is considered a primarily urban county “(61% 1967/3230 seven districts, 22 zones)” this is a bit unclear? What does 1967/3230 indicated? while Nimba County “is predominantly rural (11% (159/1471) six districts)” This is unclear? What does 159/1471 six districts? 

The sentence has been rephrased to provide more clarity. The percentage represents the communities 

3 Line 119: “The sample size was calculated using the Kish and Leslie formulae (ref).” Kindly insert ref

 This has been added.

4 Line 121: “design effect of 2” what does design effect of 2 meant? Is this part of the formula? 

The design effect of 2 indicates that the survey design has introduced some degree of clustering or stratification that increases the variance of the estimates, which means that the estimates are less precise than they would be with a more efficient design. It is not inherently part of the formular since it is a function of the sampling strategy adopted.

5 Line 178: “multiple logistic regression” What the authors did was simple binary logistic regression and should be corrected as such. 

This has now been corrected.

6 Line 200: “SMS” write in full 

The acronym has been written in full, thank for the comment 

7 Line 210-212: “When considering the effect of other variables in the model, respondents aged more than 55 years (aOR:0.5; 95% CI: 0.2–0.9; p=0.043) were two times more likely to be COVID-19 vaccination hesitant compared to those aged 45–54 years” This statement is at variance and may confuse the reader. It is ether you adjust your reference in the Table for positive odds ratio as reported in the prose or you interpreted as stated in Table 2 in lesser effects. 

The sentence has now been rephrased as in the table to limit the risk of misunderstanding.

8 Line 224: Table 2. “Multivariate association” This is multivariable and not multivariate 

The correction has been effected.

9 Lines 271-273: “With regards to age, 15-24 year-olds were more likely to express hesitancy to take the COVID-19 vaccine because the disease is more lethal to persons older and thus they did not see themselves at risk of getting the disease.” Your findings do not support this assertion/discussion. Indeed, it is contrary as vaccine hesitancy was more in those aged more than 55 years! Check your Table 2 results. 

This has been revised to align better with the result. It now reads, “Those aged more than 55 years were more likely to be hesitant about COVID-19 vaccine uptake. This may be due to limited information about the vaccine or some myth about the vaccine. Being a new disease, the younger individuals who are more on social media are more likely to access more information about the disease and may be more accepting of the vaccine. Other studies have shown that younger age groups were however more hesitant as compared to the elderly (28).”

10 Line 276: “statistically significant predictor of the hesitancy of the COVID-19 vaccine” kindly replace with association- a cross sectional data can not give you a predictor. You may wish to read more about predictive models. 

This has been updated 

11 Line 278: “Urban dwellers were usually more likely to have mobile phones and engage in social media and more likely to obtain information from other sources” This statement is hanging? Not link with the predecessor and how does it lead to VH?

Thanks for the comment, the sentence has been improved on line 288 and 289 

12 Line 280: “Our study is not without limitations. Our sample size was small and included only two of Liberia’s fifteen counties. Nevertheless, these counties accounted for 83% of all reported COVID-19 cases in the country” These statements are not limitation. Your sample size of 800 plus is adequately powered and sampling two major districts comprising of urban and rural areas out of 15 should provided an adequate representation and can not be a study limitation. Your study limitations may be lack of in-depth analysis of reasons for VH, which will require a qualitative study approach and also lack of follow up to see how many people will eventually get vaccinated.

Thanks for your observation, this section has been revised as suggested. It now reads, “Our study is not without limitations. A qualitative arm would have helped collect in-depth information to help understand other possible reasons for the hesitancy observed. Being only quantitative in nature, our study was not able to explore these. Also, we were not able to follow up with the participants to determine those who eventually took the vaccine.”

13 Line 282: “The survey provides information needed to strengthen and increase the impact of future planned mass vaccination campaigns” Move to the end of conclusion 

The line has been added to the conclusion 

14 Overall while I understood time has caught up with your data, kindly compared your key findings with other African studies e.g Babatunde Oluwatosin Ogunbosi et al. COVID-19 vaccine hesitancy in six geopolitical zones in Nigeria: a cross-sectional survey. Pan African Medical Journal. 2022;42(179). 10.11604/pamj.2022.42.179.34135 

Thanks for providing this resource document, it was compared with my results in the discussion section

Reviewer # 4 comments

1. Regarding your conclusion that vaccine hesitancy should be combated through additional information on television and radio since those are the most common sources of information at 53% - this is probably correct but I was not 100% convinced that it was supported by the data presented. Your denominator was the entire study population and I didn't see a breakdown by those who were accepting of vaccination and those who were not. 

Thanks for your comment those that were hesitant of taking the vaccine were 29.1 % (255/877). 

2. Lines 129-130: "...communities were selected based on the number of communities and their distribution in each county." I couldn't really tell from this statement what your selection criteria were. Were you looking for a greater number of communities? What type of distribution did you want, and why? 

This has been rephrased for clarity. It now reads, “From each stratum, communities were selected using simple random sampling. The number of communities selected was proportionate number of communities in each stratum in each county. Twenty-five communities were selected in all. Finally, seventeen (seven rural and ten urban) communities were selected in Montserrado; and eight (seven rural and one urban) were selected in Nimba using simple random sampling.”

3. 133-141: I found the description of Stage 2 a little hard to follow and still don't understand how you used the sampling interval to select the second household. Did you randomly select a direction again? 

Thanks for the comment more information has been provided in the stage 2 of the sampling technique for clarity.

4. General: I saw a previous reviewer recommended removing the description about street interviews. I am not sure I agree with this if street interviews formed part of your data. The questions I would ask myself are whether they formed a meaningful proportion of your data, and if you can describe it in a way that addresses the comment about it sounding haphazard. 

The data for the street interview was removed to be analyzed with other data collected during the survey. Thanks for the interest in more of our findings.

5. 186-187: There was another reviewer who asked about the ethics committee and I also had a question about this. Can you confirm that submitting to the regular ethics committee that you normally would outside a public health emergency was not an option? My concern is that an ethics committee formed of leadership from MOH and NPHIL is not what I would expect in the ERC constitution. For example there may not be diverse backgrounds and no lay person on the committee. 

Submitting to the regular ethics committee was not an option at the time of the response. Outside of the response, all research is submitted to the IRB for clearance. The leadership of both institutions consisted of the leadership and members of the Incident Management System which were multi-discipline.

6. 95: Self-employed is not listed in Table 1, only employed and unemployed. 

For the purpose of the analysis were group employment status into employed and unemployed. Self employed was merged with employed. Our interest was to check if having a stable source of livelihood is a factor and not the categories of employment. 

7. 194-196: To say most participants were unmarried (41.4%) seems a little misleading considering married participants were 38.1% and co-habiting 15.6%. 

Thank you for your comment The statement has been rephrased.

8. 210-212: Here it says >55 years is more likely to be hesitant than 45-54 but in the abstract you said the opposite. 

This has now been rephrased for consistency

9. Table 2: Interesting that the most hesitancy was seen in the 15-24 age group, but this is not mentioned in the results. 

The information has been added to the prose in the results. This was not emphasized because the association was not statistically significant

10. Table 2: I think your percentages are reversed for hesitancy yes/no in Montserrado County? 

Thanks for the comment the percentages have been recalculated and updated in table 2 

11. Figure 1: This is where I think it might be helpful to stratify by people who are vaccine hesitant those who are not. Is there a difference between those two groups in sources of information? 

The data was collected for those who were hesitant to understand the reason they are hesitant so the suggestion may not be feasible at this time. We agree it would be interesting to see the difference but unable to report that at this time

12. 254-256: Not sure this is supported by the data? In Figure 1, Facebook as a source of information seems pretty similar between urban and rural. 

The information was reported as received. Though the source information might be similar the quality of information received and their understanding of the information could be different

. 

13. 278-279: Again, I am not seeing this in Figure 1. In fact, it appears that more rural respondents relied on "other" sources of information 

The rural communities have other sources of information like town hall meetings, town crier but majority also have access to radio etc. Unfortunately, it was not separated during the data collection

---

## [Decision Letter · Decision Letter 3]

8 Nov 2023

PONE-D-22-17733R3COVID-19 Vaccine Hesitancy among adults in Liberia, April – May 2021PLOS ONE

Dear Dr. Sanvee-Blebo,

Thank you for submitting your manuscript to PLOS ONE. After careful consideration, we feel that it has merit but does not fully meet PLOS ONE’s publication criteria as it currently stands. Therefore, we invite you to submit a revised version of the manuscript that addresses the points raised during the review process.

We look forward to receiving your revised manuscript.

Kind regards,

Omar Enzo Santangelo

Academic Editor

PLOS ONE

Journal Requirements:

Reviewers' comments:

Reviewer's Responses to Questions

**Comments to the Author**

1. If the authors have adequately addressed your comments raised in a previous round of review and you feel that this manuscript is now acceptable for publication, you may indicate that here to bypass the “Comments to the Author” section, enter your conflict of interest statement in the “Confidential to Editor” section, and submit your "Accept" recommendation.

Reviewer #3: All comments have been addressed

Reviewer #4: (No Response)

2. Is the manuscript technically sound, and do the data support the conclusions?

Reviewer #3: Yes

Reviewer #4: Partly

3. Has the statistical analysis been performed appropriately and rigorously? 

Reviewer #3: Yes

Reviewer #4: I Don't Know

4. Have the authors made all data underlying the findings in their manuscript fully available?

Reviewer #3: Yes

Reviewer #4: Yes

5. Is the manuscript presented in an intelligible fashion and written in standard English?

Reviewer #3: Yes

Reviewer #4: Yes

6. Review Comments to the Author

Reviewer #3: The authors have addressed all the issues raised in my earlier review satisfactorily. I wish them all the best.

Reviewer #4: It seems that comment #1 was misunderstood. Of the 53% who got their information from television and radio, you don't state the breakdown between people who accept the vaccine and those who are hesitant. If all of the people who are hesitant got their information from other sources, it does not seem like the conclusion to focus more on television and radio would be warranted. Since this is one of your primary conclusions, it would be nice to understand this better.

7. PLOS authors have the option to publish the peer review history of their article (what does this mean?). If published, this will include your full peer review and any attached files.

Reviewer #3: **Yes: **Dr Olayinka Ibrahim

Reviewer #4: No

---

## [Author Response · Author response to Decision Letter 3]

21 Dec 2023

Thank you for the clarification. We have now conducted further analysis on the source of information which showed that the relationship between vaccine hesitancy status and receiving information through the media (Television/radio) was not found to be statistically significant (p=0.0525). Based on this new finding we have now revised the conclusion of the study to remove attention from media. We appreciate this feedback and do hope that the work is now in an acceptable format. Thank you.

---

## [Editor Report · Decision Letter 4]

27 Dec 2023

COVID-19 Vaccine Hesitancy among adults in Liberia, April – May 2021

PONE-D-22-17733R4

Dear Dr. Sanvee-Blebo,

We’re pleased to inform you that your manuscript has been judged scientifically suitable for publication and will be formally accepted for publication once it meets all outstanding technical requirements.

Kind regards,

Omar Enzo Santangelo

Academic Editor

PLOS ONE

---

## [Editor Report · Acceptance letter]

26 Mar 2024

PONE-D-22-17733R4 

PLOS ONE

Dear Dr. Sanvee-Blebo, 

I'm pleased to inform you that your manuscript has been deemed suitable for publication in PLOS ONE. Congratulations! Your manuscript is now being handed over to our production team.

Kind regards, 

on behalf of

Dr. Omar Enzo Santangelo 

Academic Editor

PLOS ONE